# Preparation of Magnetic CuFe_2_O_4_@Ag@ZIF-8 Nanocomposites with Highly Catalytic Activity Based on Cellulose Nanocrystals

**DOI:** 10.3390/molecules25010124

**Published:** 2019-12-28

**Authors:** Sufeng Zhang, Yongshe Xu, Dongyan Zhao, Wenqiang Chen, Hao Li, Chen Hou

**Affiliations:** 1College of Bioresources Chemical and Materials Engineering, Shaanxi University of Science and Technology, Shaanxi Provincial Key Laboratory of Papermaking Technology and Specialty Paper Development, Key Laboratory of Paper Based Functional Materials of China National Light Industry, National Demonstration Center for Experimental Light Chemistry Engineering Education, Xi’an 710021, China; zgyczbzgs@126.com (Y.X.); 15991346475@163.com (D.Z.); cwq1801075@163.com (W.C.); lihao120733@163.com (H.L.); houchen@sust.edu.cn (C.H.); 2Tianjin China Banknote Paper Co., Ltd., Tianjin 300385, China

**Keywords:** CuFe_2_O_4_ nanoparticles, cellulose nanocrystals, Ag nanoparticles, ZIF-8 catalysis, 4-nitrophenol reduction

## Abstract

A facile approach was successfully developed for synthesis of cellulose nanocrystals (CNC)-supported magnetic CuFe_2_O_4_@Ag@ZIF-8 nanospheres which consist of a paramagnetic CuFe_2_O_4_@Ag core and porous ZIF-8 shell. The CuFe_2_O_4_ nanoparticles (NPs) were first prepared in the presence of CNC and dispersant. Ag NPs were then deposited on the CuFe_2_O_4_/CNC composites via an in situ reduction directed by dopamine polymerization (PDA). The CuFe_2_O_4_/CNC@Ag@ZIF-8 nanocomposite was characterized by TEM, FTIR, XRD, N_2_ adsorption-desorption isotherms, VSM, and XPS. Catalytic studies showed that the CuFe_2_O_4_/CNC@Ag@ZIF-8 catalyst had much higher catalytic activity than CuFe_2_O_4_@Ag catalyst with the rate constant of 0.64 min^−1^. Because of the integration of ZIF-8 with CuFe_2_O_4_/CNC@Ag that combines the advantaged of each component, the nanocomposites were demonstrated to have an enhanced catalytic activity in heterogeneous catalysis. Therefore, these results demonstrate a new method for the fabrication of CNC-supported magnetic core-shell catalysts, which display great potential for application in biocatalysis and environmental chemistry.

## 1. Introduction

Functional nanocomposites represent an important class of nanomaterials and have attracted increased research interest due to their superior properties compared with individual components [1]. As an important member of nanocomposites family, the magnetic nanocatalyst is very attractive because it provides a convenient way to remove and recycle the nanocatalyst from the reaction system [2]. MFe_2_O_4_ ferrite, which is a well-known ternary spinel structure with M^2+^ ions on B sites and Fe^3+^ ions located equally among A and B sites, has high thermal, mechanical, and chemical stability and versatile catalytic, electric, and magnetic properties, and it exhibits promise for applications in electronics, lithium ion batteries, sensors, catalysis, and diagnostic medicine [3,4]. CuFe_2_O_4_ possesses high electronic conductivity, high thermal stability, and high activity for the hydrogenation of 4-nitrophenol (4-NP), which is a toxic and inhibitory in nature, to yield industrially important anilines like 4-aminophenol [5,6,7].

More often, hybrid composites are fabricated through an in-situ method, where the templates have a high affinity towards metal ions that allows for the synthesis of metallic nanoparticles [8]. Cellulose nanocrystals (CNC) are derived from abundant cellulosic resources such as plants and microbial cellulose via sulfuric acid hydrolysis [9,10]. CNC have well-defined size and morphology, high specific surface area, high aspect ratio, high crystalline order, chirality, high mechanical strength, and controllable surface chemistry [11], which are appealing in a plethora of materials to catalytic applications. As reported in our previous research [12], the CuFe_2_O_4_/CNC nanocomposites show good dispersity and it has been suggested that the nanocomposites do catalyze 4-NP reduction.

To further improve the catalytic activity, various core/shell-structured magnetic nanocatalysts have been applied in nanocatalysis [13,14,15,16]. The as-obtained CuFe_2_O_4_ based core-shell nanocomposites showed excellent catalytic activity, magnetic separation, and magnetic carrying in nanocatalysis. Recently, the Ag modified magnetic composites have gained increasing attention because of the high catalytic activity of the Ag component, the good magnetic responsiveness of the magnetic core, and the relatively facile fabrication process [17,18,19]. Various noble metals, including Au, Pd, and Pt have been widely employed as catalysts for the reduction of 4-NP to 4-AP by NaBH_4_ in aqueous media [20,21,22,23]. Moreover, Ag is more suitable for large-scale application than other noble metals (Au, Pd, and Pt) because of its low price [24]. Dopamine can self-polymerize under specific conditions to form a polydopamine (PDA) complex which has the ability of adhering onto the surface of various materials due to the strong stickiness [25,26,27] and the abundant catechol groups on the PDA complex can reduce metal into metal directly [28,29]. In this regard, Ag catalysts supported on ferrite, Fe_3_O_4_, Fe_3_O_4_@PDA, etc., have been used as magnetic catalysts for catalytic reduction. Recently, Wu et, al. fabricated Fe_3_O_4_@PDA-Ag nanocomposites and used them as an efficient catalyst for methylene blue reduction owing to Ag nanoparticles (NPs) [27]. CNC, crystalline cellulose nanofibers, and their derivatives, were good supports for the preparation of supported Ag NPs [11,30]. Moreover, CNC have eminent colloidal stability due to the interelectrostatic repulsions of the negatively charged sulfate ester groups on CNC [31,32]. These sulfate groups have been proven to favor stabilization and nucleation of Ag NPs in a recent research [11]. However, the introduction of CNC in magnetic catalyst is uncommon. The catalyst with CNC combines the advantages of magnetic and catalytic.

The ZIF-8 framework (Zn(MeIM)_2_, MeIM = 2-methylimidazole) holds an intersecting 3D structure, high thermal and chemical stabilities, and large pore size and surface area, which are desirable for depositing metal NPs [33]; it can also rapid adsorb chemical pollutants from aqueous solutions. ZIF-8 has uniform but tunable cavities, tailorable chemistry, and is suited to stabilizing noble metal-NPs without blocking their surfaces, making them very attractive in catalysis [7]. Furthermore, proper design and construction of ZIF-8 nanocomposites by integrating functional materials (such as metal NPs, metal oxides, and polymers) are expected to present synergistically boosted catalytic activity, enhanced stability, and prolonged lifetimes [34,35] which would open new opportunity for fabricating highly efficient hierarchical core-shell catalysts [36].

Herein, we report the synthesis of cellulose nanocrystals (CNC) supported magnetic CuFe_2_O_4_@Ag@ZIF-8 nanocomposites. It consists of a paramagnetic CuFe_2_O_4_/CNC @Ag core and a porous ZIF-8 shell via a facile method. CNC was used as a template and dispersant for the incorporate with CuFe_2_O_4_ NPs and an absorbent via π–π stacking interactions of 4-NP. In addition, we used PDA as an intermediate, which can not only protect the CuFe_2_O_4_ NPs from corrosion in harsh environments and immobilize Ag NPs, but also induce the growth of ZIF-8 on the surface of the CuFe_2_O_4_/CNC@Ag. The as-synthesized CuFe_2_O_4_/CNC@Ag@ZIF-8 nanocomposites showed outstanding catalytic activity and reusability for the reduction of 4-NP. To our best knowledge, this is the first report on the integration of ZIF-8 with CuFe_2_O_4_/CNC@Ag into one nanostructure to significantly improve not only the intrinsic catalytic performances of Ag active species, but also the recyclability of catalysts.

## 2. Results and Discussion

### 2.1. Characterization of CuFe_2_O_4_/CNC@Ag and CuFe_2_O_4_/CNC@Ag@ZIF-8 Nanocomposites

The general schemes for the synthesis of the CuFe_2_O_4_/CNC@Ag@ZIF-8 nanocomposites are illustrated in Scheme 1, which mainly involved three steps: (1) Firstly, one-pot solvothermal synthesis of cellulose nanocrystals supports CuFe_2_O_4_ NPs, and the network of cellulose nanocrystals can significantly enhance the dispersion stability. (2) Secondly, as a result of its unique self-adhesive and reductive nature, the PDA layer shows sufficient reductive capacity to reduce Ag^+^ ions without the need for additional reducing agent [37]. By then, Ag NPs with sizes ranging from 20 to 30 nm in sphere shapes can be uniformly formed along the CuFe_2_O_4_ and CNC networks surface with the aid of adhesive and reductive PDA layer under alkaline conditions. (3) Thirdly, using Zn(NO_3_)_2_·2H_2_O and 2-methylimidazole as precursors, the ZIF-8 shell with controllable thickness was coated on the surface of CuFe_2_O_4_/CNC@Ag. The composite is prepared by layer-by-layer.

Transmission electron microscopy (TEM) measurements revealed that the CuFe_2_O_4_ NPs had good monodispersity in the CNC network with a mean size of about 250 nm (Figure 1A). The pristine CNC) had a length of ca. 200–250 nm and a width of 15–20 nm, which is typical for CNC [38]. CuFe_2_O_4_ NPs shown in Figure 1A were well dispersed in the presence of CNC substrate, which supported the conclusion that CNC can act as a good dispersant/support of nanoparticles, preventing the NPs’ aggregation due to the excellent properties of CNC [38]. Figure 1B showed that uniform and dense Ag NPs, with an average size of 25 nm successfully distributed on the PDA surface, and no free Ag NPs were observed. The CNC networks became dark after coating PDA. The thin PDA shell layers formed around the CuFe_2_O_4_/CNC cores showed an average size of about 20 nm (Figure 1C), displaying a distant core-shell structure. Figure 1E confirmed that the boundary between the ZIF-8 and PDA was obscure, which was attributed to the slight mass difference of the two components. Compared with CuFe_2_O_4_/CNC@Ag, CuFe_2_O_4_/CNC@Ag@ZIF-8 nanocomposites had a distinct core-shell structure; the thickness of ZIF-8 shell surrounding the CuFe_2_O_4_/CNC@Ag was approximately 45 nm, and there was a controllable particle diameter ranging from 350 to 400 nm. Figure 1F displays a lattice resolved HRTEM image of CuFe_2_O_4_ nanocrystal on CNC. The distinct lattice fringes with interplanar spacings of 0.25 and 0.235 nm match well the (311) crystal plane of the CuFe_2_O_4_ cubic spinel structure [39] and the (111) plane of Ag, respectively [40].

HAADF-STEM also confirmed the typical nanostructure of the nanocomposites (Figure 2A). The EDX mapping of the Cu and Fe elements revealed that CuFe_2_O_4_ was mainly located within the nanocomposites (Figure 2B). The diameters of the N element map were larger than that of the Cu and Fe, which further supported the fact that the PDA were successfully coated. The Ag element was distributed around the CuFe_2_O_4_ because the density in the center was very low. In addition, the corresponding EDX spectra supported the conclusion that Ag NPs were embedded in the CNC substrate [38], forming the CuFe_2_O_4_/CNC@Ag@ZIF-8 nanocomposites. The diameters of the Zn element map were larger than that of the Ag, which further supported the fact that the Ag NPs were protected by ZIF-8. The Ag element was distributed on C element, which confirmed that Ag NPs were located on PDA shell (Figure 2C–I). On the basis of the TEM and EDX mapping, it could be concluded that the CuFe_2_O_4_/CNC@Ag@ZIF-8 nanocomposites with core-shell structure had been successfully achieved in Figure 2J.

Figure 3A presents the FTIR spectra of (a) CuFe_2_O_4_/CNC, (b) CuFe_2_O_4_/CNC@PDA, (c) CuFe_2_O_4_/CNC@Ag, and (d) CuFe_2_O_4_/CNC@Ag@ZIF-8. In line (a), the adsorption peaks at 430 cm^−1^, 580 cm^−1^, and 3425 cm^−1^ correspond to the Fe–O, Cu–O, and O–H stretching vibrations, respectively [41,42]. The band at 1645 cm^−1^ was due to the O–H bending vibration in CNC [43]. The absorbance bands at 2893 cm^−1^, 1400 cm^−1^ and 1060 cm^−1^ were assigned to the C–H stretching vibration, the C–H deformation vibration and the C–O–C stretching of pyranose, respectively [15,44], which indicated that the CuFe_2_O_4_ NPs were successfully immobilized on the CNC. In line (b), besides the characteristic adsorption peaks of line (a), the adsorption peaks at 1513 cm^−1^ are related to the C=C stretching vibrations of aromatic ring [29]; the broad peak at 3390 cm^−1^ is attributed to the O–H and N–H stretching vibrations [42]; the peak appearing at 1294 cm^−1^ can be assigned to the C–OH stretching vibration of phenol compounds [43]; all the peaks above demonstrate that the CuFe_2_O_4_/CNC@PDA nanocomposites were successfully prepared. After immobilizing Ag on the CuFe_2_O_4_/CNC@PDA (line c), the intensity of the peak at 1294 cm^−1^ become weaker owing to the interaction between Ag NPs and PDA. The band at 421 cm^−1^ (shown in line (d)) was attributed to the Zn–N stretch mode [44]. The bands in the spectral region of 500–1350 cm^−1^ and 1350–1500 cm^−1^ were assigned as the plane bending and stretching of imidazole ring, respectively [44]. The bands of 2500–3500 cm^−1^ could be ascribed to stretching vibrations of –CH_3_, –NH– and –OH (Zn–OH) within the internal structure of ZIF-8 [45].

To probe the presence of Ag NPs and ZIF-8 attached onto the CuFe_2_O_4_/CNC, XRD patterns were carried out during the experiments (Figure 3B). It was observed that for CuFe_2_O_4_/CNC, two peaks at 2θ = 11.27 and 21.94° corresponded to the typical (101) and (020) lattice planes of cellulose [37], and the diffraction peaks located at 18.5°, 30.2°, 35.5°, 37.0°, 43.4°, 57.3°, and 62.6° corresponded to the (111), (220), (311), (222), (400), (422), (511), and (440) lattice planes, which matched well with those from the JCPDS card number 25-0283 for CuFe_2_O_4_ [46]. But for CuFe_2_O_4_/CNC@Ag and CuFe_2_O_4_/CNC@Ag@ZIF-8 nanocomposites, the XRD diffraction peaks derived from cellulose showed a slight decrease, and meanwhile, four diffraction peaks at 2θ = 39.86°, 44.23°, 64.47°, and 77.33° appeared, assigned respectively, to (111), (200), (220), and (311) lattice planes, and supported the face-centered cubic (fcc) structure of Ag NPs [47,48], an indication of successful formation of Ag NPs via efficient in situ reduction by PDA layer. The XRD pattern of CuFe_2_O_4_/CNC@Ag@ZIF-8 nanocomposites indicated that the products were well crystallized and had high crystallinity even after coating ZIF-8 shell. Moreover, the diffraction peaks at 2θ = 10.4°, 12.8°, 14.7°, 16.5°, and 18.1° correspond to the (002), (112), (022), (013), and (222) lattice planes of ZIF-8 in the CuFe_2_O_4_/CNC@Ag@ZIF-8 nanocomposites, respectively, suggesting that the ZIF-8 materials synthesized using current protocol are highly crystalline.

The porosity of evacuated composites was investigated by nitrogen-sorption measurements. As presented in Figure 4A, the CuFe_2_O_4_/CNC@Ag and CuFe_2_O_4_/CNC@Ag@ZIF-8 nanocomposites exhibited a typical type V isotherm, validating a mesoporous characteristic [7]. The pore-size distribution (Figure B) revealed that the CuFe_2_O_4_/CNC@Ag@ZIF-8 nanocomposites contained an average pore size of 4.0 nm which is lower than that of CuFe_2_O_4_/CNC@Ag (8.9 nm) in favor of the prevention of Ag active sites leaching. In addition, the specific surface area and the pore volume of the CuFe_2_O_4_/CNC@Ag@ZIF-8 nanocomposites were calculated to be 160.17 m^2^/g, which is four times higher than that of CuFe_2_O_4_/CNC@Ag (38.68 m^2^/g). The high external surface area and mesoporous structure endowed the CuFe_2_O_4_/CNC@Ag@ZIF-8 nanocomposites with high adsorption capacity and fast diffusion of reactants [17].

The saturation magnetization (*M*_s_) is a physical quantity that can reflect the magnetism of a substance. Both CuFe_2_O_4_/CNC@Ag and CuFe_2_O_4_/CNC@Ag@ZIF-8 nanocomposites were paramagnetic with little hysteresis and remanence, processing *M*_s_ values of 31.2 and 30.1 emu/g, respectively. Due to the coating of antimagnetic ZIF-8 shell, the saturation magnetization (*M*_s_) decreased a little. As presented in Figure 4B, the CuFe_2_O_4_/CNC@Ag@ZIF-8 nanocomposites were well dispersed in water and presented a black suspension. However, rapid aggregation (≈1 min) of the CuFe_2_O_4_/CNC@Ag@ZIF-8 nanocomposites from the homogeneous suspension was obtained with the help of external magnet, and thus the dispersed solution became clear.

### 2.2. Catalytic Reduction of 4-Nitrophenol

Ag NPs have been generally used as excellent catalysts with high catalytic activity and selectivity for catalytic reduction or degradation of organic pollution in aqueous solution [38,39,46,49]. Many reports are available on the application of metal and metal oxides nanocatalysts for the reduction of nitrophenols in the presence of NaBH_4_ [50]. Herein, the catalytic reduction of 4-NP by NaBH_4_ was used as a model reaction to investigate the catalytic performances of CuFe_2_O_4_/CNC, CuFe_2_O_4_/CNC@Ag, and CuFe_2_O_4_/CNC@Ag@ZIF-8 nanocomposites. Although the aqueous solution of 4-nitrophenol undergoes a rapid color change (with a UV–Vis absorption peak shift from 319 to 400 nm) after adding NaBH_4_ due to the formation of 4-nitrophenolate ions, the reduction reaction does not proceed substantially in the absence of suitable catalysts (such as Au, Ag, Pd, and Pt NPs).

Figure 5A suggested the catalytic reaction of CuFe_2_O_4_/CNC@Ag nanocomposites could be completed within 11 min. As shown in Figure 5B, the adsorption peak at 400 nm was observed to decrease in intensity rapidly and disappear eventually after 6 min, suggesting that the CuFe_2_O_4_/CNC@Ag@ZIF-8 nanocomposites do catalyze 4-NP reduction. Since the ZIF-8 itself, in the control experiment, exhibited no propensity to catalyze the reduction reaction, the above result indicates that the 4-nitrophenol molecules can diffuse quickly through the channels of ZIF-8 matrix and react on the surfaces of the active CuFe_2_O_4_@Ag NPs. In addition, the appearance of the new peak at ≈300 nm in the UV–Vis spectra suggested that the CuFe_2_O_4_/CNC@Ag@ZIF-8 nanocomposites catalyze the reduction of 4-NP to give 4-aminophenol as the sole product.

To elucidate the reaction mechanism, the concentration of NaBH_4_ could be considered as constant throughout the reaction since it was in great excess (0.1 M). Therefore, pseudo-first-order kinetics with regard to the catalytic reduction of 4-NP, described as ln(*C*_t_/*C*_0_) = −*kt*, can be applied, where *C*_t_ is the concentration of 4-NP at time *t*, *C*_0_ is the initial concentration of 4-NP, and *k* is the rate constant {51}. Figure 5C shows the linear relationship of ln(*C*_t_/*C*_0_) as a function of reaction time *t* for the 4-NP reduction catalyzed by three as-synthesized catalysts. The values of kinetic rate constant *k* can be calculated from the rate equation ln(*C*_t_/*C*_0_) = −*kt*. Impressively, the CuFe_2_O_4_/CNC@Ag@ZIF-8 nanocomposites exhibit the highest activity with a rate constant estimated to be 0.64 min^−1^,.5 and eight times higher than that of CuFe_2_O_4_/CNC@Ag and that of CuFe_2_O_4_/CNC, respectively, suggesting the higher catalytic efficiencies for Ag catalysts confined in ZIF-8 shell. This prominent catalytic activity can due to a porous ZIF-8 shell for stabilization of the encapsulated Ag NPs and rapid adsorption of chemical pollutants from aqueous solution. The catalytic active sites are both CuFe_2_O_4_ and Ag NPs in the core-shell structures, which modified the electronic structure, and then enhanced the catalytic activity.

The reusability of CuFe_2_O_4_/CNC@Ag@ZIF-8 nanocomposites as the catalyst for the reduction of 4-NP was further confirmed by the observation of the similar conversion for the same reaction time (6 min) for six consecutive cycles (the slightly decreased conversions in the later catalysis cycles were presumably caused by the loss of catalyst during the washing process between cycles (Figure 5D).

### 2.3. Reaction Mechanism of CuFe_2_O_4_/CNC@Ag@ZIF-8 Nanocomposites

As illustrated in Scheme 2, the mechanism of catalytic reduction of 4-NP by the CuFe_2_O_4_/CNC@Ag@ZIF-8 nanocomposites involved the traditional theory. In this work, the 4-NP can be adsorbed onto the mesoporous ZIF-8 shell via π–π stacking interactions because 4-NP is π-rich in nature [51]. Such chemical adsorption provides a high concentration of 4-NP near to the interface of the CuFe_2_O_4_@Ag and ZIF-8, leading to highly efficient contact between them. Simultaneously, ${\mathrm{BH}}_{4}^{-}$ was also adhered to the CuFe_2_O_4_/CNC@Ag surface and transferred electrons and hydride ions to the Ag NPs’ surface. In addition, Zhou et al. claimed that when metal oxide closely contacted with metal, Fermi level alignment would lead to charge redistribution: electrons would escape from the metal and transfer into the semiconductor [52]. Liang et, al. used Ag/Fe_3_O_4_ NPs as the catalyst for the reduction of 4-NP [53]. They reported that the electrons tended to leave Ag to Fe_3_O_4_ and thus form a depleted region close to the Ag/Fe_3_O_4_ interface. In our work, CuFe_2_O_4_ was known as a p-type semiconductor with low band gap, so part of the electrons and hydride ion that injected from ${\mathrm{BH}}_{4}^{-}$ to Ag NPs could transfer to the neighboring CuFe_2_O_4_ surface (Scheme 2). The existence of the surplus electrons on CuFe_2_O_4_ provided large surface area and increased opportunities for reduction reaction, facilitating the capture of electrons by 4-NP molecules. Goyal et al. suggested that electron transfer between Cu^+^-Cu^2+^ and Fe^2+^-Fe^3+^ in the octahedral sites endowed CuFe_2_O_4_ with enhanced catalytic activity [54] (Scheme 2). Herein, when the electrons and hydride ions were transfered to the CuFe_2_O_4_ surface, both Cu^2+^and Fe^3+^ ions present in the octahedral sites were exposed on the surfaces of particles. Due to that, there were transfers of electrons between Cu^+^-Cu^2+^ and Fe^2+^-Fe^3+^ ion pairs, which enhanced catalytic activity. Then the hydrogen atom transfers from ${\mathrm{BH}}_{4}^{-}$ to the 4-NP, resulting in the formation of 4-AP. Finally, the products of 4-AP are desorbed from the surface of the catalysts to the solution through the channels of the ZIF-8 shell.

It should be noted that the bleaching rate is considerably higher than the rates reported previously under the similar experimental conditions with Ag-based, CuFe_2_O_4_-based, and MOF-based catalysts. The *k* values of different catalytic systems for the reduction of 4-NP were comparable to the values referenced in Table 1, and the results showed that the prepared catalyst possessed higher activity and lower activation energy.

## 3. Materials and Methods

### 3.1. Chemicals

All the reagents used were of analytical purity and were used without further purification. Polyethylene glycol-6000 (PEG-6000), polyvinylpyrrolidone (PVP-1300000), dopamine hydrochloride (DA·HCl), and 2-methylimidazole (C_4_H_6_N_2_) were purchased from Shanghai Aladdin Industrial Co., Ltd. (Shanghai, China). Cupric chloride anhydrous (CuCl_2_) and other reagents, such as ferric chloride hexahydrate (FeCl_3_·6H_2_O), ammonium acetate (NH_4_OAc), ethylene glycol (EG), silver nitrate (AgNO_3_), 4-nitrophenol(4-NP), sodium borohydride (NaBH_4_), and zinc nitrate (Zn(NO_3_)_2_·6H_2_O) were purchased from Sigma-Aldrich (Shanghai, China). Cellulose nanocrystals (CNCs) were purchased from Tianjin Haojia Cellulose Co., Ltd. (Tianjin, China).

### 3.2. Measurements

Sample morphologies with energy-dispersive X-ray spectroscopy (EDX) were characterized by transmission electron microscopy (TEM) on a TECNAI G2 TF20 (U.S.). FT-IR spectra of all samples in the wavenumber range 4000–400 cm^−1^ were obtained in KBr pressed pellets on a TENSOR model 27 FTIR spectrometer (Germany, Bruker). The powder X-ray diffraction spectra (XRD) were measured by X-ray diffraction (Germany, Bruker, D8Advance) with Cu Kα radiation, λ = 1.542 Å. The specific surface area was calculated by the Bruner–Emmett–Teller (BET) method. The pore size distributions were derived from the adsorption branches of the isotherms based on the Barrett–Joyner–Hollande (BJH) model. Magnetic hysteresis loops at room temperature were obtained using a vibrating sample magnetometer VSM 7304 (Lakeshore, Columbus, OH, USA). The chemical composition of nanocomposites was characterized by XPS (U.S. Thermos Scientific ESCALAB250). The UV–Vis spectra (China, Shanghai, Shimadzu UV-2501 PC spectrometer) were performed to study the catalytic reduction activity. The samples were placed in a 1 × 1 × 3 cm quartz cuvettes, and the spectra were recorded at room temperature.

### 3.3. Preparation of CuFe_2_O_4_/CNC Nanocomposites

In a typical preparation, the procedure was reported as per previous research [12]. CNC (0.2 g) was dispersed in 40 mL of glycol with vigorous stirring in an ultrasonic generator for 0.5 h. On the other hand, 1.6 mmol CuCl_2_·2H_2_O and 3.2 mmol FeCl_3_·6H_2_O were dissolved in 20 mL of glycol to form a clear solution. After complete dissolution, CNC solution was poured into the metal precursor solution and followed by the addition of 0.2 g PVP while stirring for 0.5 h. Addition of NH_4_OAc (90 mmol) in a stepwise manner was done to the mixture until homogeneous light green dispersion. Then, the mixture was transferred into a Teflon-lined stainless steel autoclave (80 mL capacity) and heated at 200 °C for 11 h. After the reaction, the autoclave was naturally cooled to room temperature, and the catalysts were collected and washed with redistilled water and ethanol three times, respectively. Finally, the catalysts were dried in a vacuum for 4 h at 60 °C.

### 3.4. In Situ Reduction of Ag^+^ Ions

To coat CuFe_2_O_4_/CNC nanocomposites with the PDA shell, 50 mg CuFe_2_O_4_/CNC nanocomposites and 50 mg of dopamine hydrochloride were dissolved in 25 mL Tris buffer solution (10 mM, pH = 8.5). After shaking for 3 h at room temperature, the CuFe_2_O_4_/CNC@PDA were separated and washed with ultrapure water and ethanol several times. For the preparation of Ag NPs on PDA surfaces, Tollen’s reagent (silver ammonia solution) was used as the Ag precursor solution. Silver ammonia solution was prepared by adding ammonia aqueous solution (2 wt %) into 10 mg·mL^−1^ AgNO_3_ solution until brown precipitation was just dissolved. Portions 50 mg in size of the CuFe_2_O_4_/CNC@PDA nanocomposites were added to 25 mL of silver ammonia solution, and the mixture was shaken in a rotary shaker for 6 h at room temperature. The products were collected, washed with ultrapure water and ethanol several times, and dried under vacuum. Then, CuFe_2_O_4_/CNC@Ag nanocomposites were obtained.

### 3.5. Preparation of CuFe_2_O_4_/CNC@Ag@ZIF-8 Nanocomposites

Briefly, 50 mg CuFe_2_O_4_/CNC@Ag nanocomposites were added into Zn(NO_3_)_2_·6H_2_O methanol solution (10 mL, 50 mM) and stirred for 10 min at 50 °C. Subsequently, the nanocomposites were dispersed in methanol solution (10 mL, 500 mM) under stirring for 30 min at 50 °C, collected by a magnet, cleaned with ultrapure water and ethanol, and dried under vacuum at 60 °C overnight.

### 3.6. General Procedure for the Reduction of 4-NP

The reduction of 4-NP by NaBH_4_ was chosen as a model reaction for investigating the catalytic performance of the CuFe_2_O_4_/CNC, CuFe_2_O_4_/CNC@Ag, and CuFe_2_O_4_/CNC@Ag@ZIF-8 nanocomposites. Typically, 2.35 mL ultrapure water, 200 μL 5mM 4-NP solution, and 450 μL 200 mM of fresh prepared NaBH_4_ aqueous solution were added into standard quartz cuvettes respectively, and the solutions turned bright yellow rapidly. Subsequently, 3 mg of each catalysts was added to start the reaction, and the intensity of the absorption peak at 400 nm was monitored by UV–Vis spectroscopy as a function of time.

## 4. Conclusions

In summary, we demonstrated an effective strategy for the fabrication of novel cellulose nanocrystals (CNC)-supported magnetic CuFe_2_O_4_@Ag@ZIF-8 catalysts which consist of a paramagnetic CuFe_2_O_4_@Ag core and a porous ZIF-8 shell. The use of CNC include being a template and dispersant for the incorporation with CuFe_2_O_4_ NPs and a good absorbent via π–π stacking interactions of 4-NP. The framework matrix of the resulting composites retains its high surface areas, uniform mesoporous structure, porous crystalline structure, and good magnetic response. The core-shell magnetic catalysts were found to exhibit excellent catalytic performance for 4-nitrophenol reduction with good reusability. Compared to CuFe_2_O_4_/CNC@Ag catalysts, the core-shell structure CuFe_2_O_4_/CNC@Ag@ZIF-8 nanocomposites are ideal recyclable catalysts for liquid-phase reductions due to a porous ZIF-8 shell for stabilization of the encapsulated Ag NPs and rapid adsorption of chemical pollutants from aqueous solution. More importantly, with the merits of easy separation and porous shell structure, this simple and versatile method might provide a multitude of noble, ZIF-8, and magnetic catalysts for broad applications, such as environmental protection, chemical biosensors, and so on.

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
