# Peer review of "Preparation of Magnetic CuFe2O4@Ag@ZIF-8 Nanocomposites with Highly Catalytic Activity Based on Cellulose Nanocrystals"

_molecules, 2019, doi:10.3390/molecules25010124_

Round 1

Reviewer 1 Report

The manuscript "Preparation of Magnetic CuFe2O4@Ag@ZIF8 Nanocomposites with Highly Catalytic Activity Based on Cellulose Nanocrystals" by the authors S. Zhang, Y. Xu, D. Zhao, W. Chen, H. Li, C. Hou presents synthesis of cellulose nanocrystals (CNC) supported magnetic CuFe2O4@Ag@ZIF-8 nanospheres which consist of a paramagnetic CuFe2O4@Ag core and porous ZIF-8 shell. The CuFe2O4/CNC@Ag@ZIF-8 nanocomposite was characterized by TEM, FTIR, XRD, N2 adsorption-desorption isotherms, VSM and XPS. Catalytic studies showed that the CuFe2O4/CNC@Ag@ZIF-8 catalyst had much higher catalytic activity than CuFe2O4@Ag catalyst. Because of the integration of ZIF-8 with CuFe2O4/CNC@Ag that combines the advantaged of each component, the nanocomposites were demonstrated with an enhanced catalytic activity in heterogeneous catalysis.

This is a very thorough and interesting study, the work is well performed and the paper clearly and systematically described the results, achieved by many different methods of characterization. The publication of this work in Molecules is recommended, authors should consider only one comments/suggestions:

Please check this statement:

“….MFe2O4 ferrites, which is a well-known ternary spinel structure with M2+ ions on B sites..”

Author Response

Response to Reviewer 1 Comments Point 1: Please check this statement “…MFe2O4 ferrites, which is a well-known ternary spinel structure with M2+ ions on B sites.” Response 1: We check this statement “…MFe2O4 ferrites, which is a well-known ternary spinel structure with M2+ ions on B sites and Fe3+ ions located equally among A and B sites…” carefully. We confirm the statement is cited by the reference correctly.

Reviewer 2 Report

The authors present a novel approach to synthesis of cellulose nanocrystals (CNC) supported magnetic CuFe2O4@Ag@ZIF-8 nanospheres which consist of a paramagnetic CuFe2O4@Ag core and porous ZIF-8 shell. Overall, the paper is well written, with clearly presented results and conclusion. The authors show that the addition of ZIF-8 highly increases catalytic activity of CuFe2O4 and I think that it will be interesting to the audience of the Molecules journal.

In my opinion, the part of the manuscript which is lacking is the XRD analysis. The authors claim that the existence of Ag nanoparticles is clear from the XRD peaks located at 39.86°, 44.23°, 64.47°, and 77.33°. It would be opportune to do the Rietveld refinement, or at least some sort of simpler crystallite size analysis (Warren-Averbach or Williamson-Hall) to determine the size of the Ag particles to determine whether they are really nano sized.

Furthermore, the language used in the part dealing with XRD analysis is quite strange, when stating the Bragg angles it is customary to state that they belong to hkl lattice planes, not to crystal faces so I suggest that the authors change that.

Afther these minor corrections my opinion is that the manuscript can be published in Molecules.

Author Response

Point 1: In my opinion, the part of the manuscript which is lacking is the XRD analysis. The authors claim that the existence of Ag nanoparticles is clear from the XRD peaks located at 39.86°, 44.23°, 64.47°, and 77.33°. It would be opportune to do the Rietveld refinement, or at least some sort of simpler crystallite size analysis (Warren-Averbach or Williamson-Hall) to determine the size of the Ag particles to determine whether they are really nano sized.

Response 1: The XRD analysis shows the structure and compounds of the composites. To further improve the Ag particles of the nano sized, we use the lattice resolved HRTEM image (HRTEM) to show the nanostructure of Ag particles as shown in Fig.1F. Fig. 1F displays a lattice resolved HRTEM image of Ag particles with nano sized.

Point 2: Furthermore, the language used in the part dealing with XRD analysis is quite strange, when stating the Bragg angles it is customary to state that they belong to hkl lattice planes, not to crystal faces so I suggest that the authors change that.

Response 2: We change the “crystal faces” to “lattice planes” in the part with XRD analysis, as shown in the revised paper.

Reviewer 3 Report

The manuscript describes an interesting study of magnetic, porous nanocomposites for catalytic Applications. It is difficult to evaluate the novelty and impact, as the introduction is largely lacking a section describing knowledge needs and limitations in the state-of-the-art - the manuscript should be updated accordingly.

Another shortcoming is that the nomenclature used is somewhat misleading; the use of "@" in this context often implies a core-shell structure, in this case that you start out with a CuFe2O4 core and form a silver Shell based on secondary nucleation, but this is clearly not the case. The claim that the procedure yields a ZIF-8 Shell onto the magnetic particle core also seems to be somewhat dubious e.g. from Fig 1, where the porous structure only partly overlaps With the magnetic particles. I suggest a reworking of the manuscript to better describe the structures formed.

Based on N2 adsorption-desorption isotherms, the Authors claim that the obtained nanocomposites "..exhibited a typical type IV isotherm, validating a mesoporous characteristic." While there is a slight hysteresis observed in both systems which indicates capillary condensation, only the ZIF-8 system appears to have the initial Plateau/inflection at low relative pressures which corresponds to completion of monolayer coverage corresponding to type IV With an H3 or H4 type structure, whereas the other system appears to be more consistent With a type V isotherm (also backed up by the respective Surface areas). Please update the mansucript accordingly.

Author Response

Point 1: It is difficult to evaluate the novelty and impact, as the introduction is largely lacking a section describing knowledge needs and limitations in the state-of-the-art - the manuscript should be updated accordingly.

Response 1: We update the section describing knowledge needs and limitations in the introduction of the revised paper.

Point 2: Another shortcoming is that the nomenclature used is somewhat misleading; the use of "@" in this context often implies a core-shell structure, in this case that you start out with a CuFe2O4 core and form a silver Shell based on secondary nucleation, but this is clearly not the case. The claim that the procedure yields a ZIF-8 Shell onto the magnetic particle core also seems to be somewhat dubious e.g. from Fig 1, where the porous structure only partly overlaps With the magnetic particles. I suggest a reworking of the manuscript to better describe the structures formed.

Response 2: We update the manuscript accordingly. Though the composite is not a typical core-shell structure, we use "@" to show the layer-by-layer or in-situ reduction method in preparation.

Point 3: Based on N2 adsorption-desorption isotherms, the Authors claim that the obtained nanocomposites "..exhibited a typical type IV isotherm, validating a mesoporous characteristic." While there is a slight hysteresis observed in both systems which indicates capillary condensation, only the ZIF-8 system appears to have the initial Plateau/inflection at low relative pressures which corresponds to completion of monolayer coverage corresponding to type IV With an H3 or H4 type structure, whereas the other system appears to be more consistent With a type V isotherm (also backed up by the respective Surface areas). Please update the mansucript accordingly.

Response 3: We update the manuscript accordingly.

Round 2

Reviewer 3 Report

The Authors have modified the mansucript according to reviewer comments